# Multi-Timescale Dynamics Model Bayesian Optimization for Plasma Stabilization in Tokamaks

**Rohit Sonker** [* 1]  **Alexandre Capone** [* 1]  **Andrew Rothstein** [2]  **Hiro Josep Farre Kaga** [2]  **Egemen Kolemen** [2 3]  **Jeff Schneider** [1]

## Abstract

Machine learning algorithms often struggle to control complex real-world systems. In the case of nuclear fusion, these challenges are exacerbated, as the dynamics are notoriously complex, data is poor, hardware is subject to failures, and experiments often affect dynamics beyond the experiment's duration. Existing tools like reinforcement learning, supervised learning, and Bayesian optimization address some of these challenges but fail to provide a comprehensive solution. To overcome these limitations, we present a multi-scale Bayesian optimization approach that integrates a high-frequency data-driven dynamics model with a low-frequency Gaussian process. By updating the Gaussian process between experiments, the method rapidly adapts to new data, refining the predictions of the less reliable dynamical model. We validate our approach by controlling tearing instabilities in the DIII-D nuclear fusion plant. Offline testing on historical data shows that our method significantly outperforms several baselines. Results on live experiments on the DIII-D tokamak, conducted under high-performance plasma scenarios prone to instabilities, shows a 50% success rate — marking a 117% improvement over historical outcomes.

## 1. Introduction

Controlling real-world systems is inherently challenging, even when powerful machine-learning tools are employed: nonlinearities are often pronounced, data is scarce, and safety issues impose severe limitations. These challenges are especially pronounced when controlling tokamaks, a form of nuclear fusion reactor. For tokamaks, good models are unavailable, safety is paramount, and instabilities are notoriously hard to control. These issues are further complicated by the fact that the dynamics may fluctuate strongly due to various reasons: tokamaks experience dozens of experiments on any given day, some of which deliberately change its base configurations, e.g., by introducing impurities into the plasma, potentially affecting the dynamics of posterior experiments. Moreover, tokamaks frequently undergo hardware changes or outages, e.g., wall repairs or beam failures, making any single specific model unreliable. However, despite these challenges, designing good control policies for tokamaks is highly desirable due to their promise to generate abundant clean energy via nuclear fusion.

In many real-world control applications, model-free reinforcement learning is a promising solution and has seen successful applications (He et al., 2024; Kumar et al., 2021; Lee et al., 2020). However, most of these methods rely on a prohibitive amount of policy rollouts for training, which is typically only achievable with reliable simulation environments. In complex environments like tokamaks, this is particularly problematic, as operation costs typically only permit a handful of rollouts, and existing simulators do not reflect the true dynamics for many aspects of the plasma (Char et al., 2023a). Offline RL seeks to overcome these issues by learning a policy from offline data that conservatively stays within the bounds of the observed data (Levine et al., 2020). However, the performance of offline RL methods depends crucially on high-quality expert data that contains advantageous states. If these are not present, then offline RL can suffer from extrapolation errors (Fujimoto et al., 2019). This is a major drawback for tokamak control, where significant exploration and improvement are still required to achieve energy production.

Alternatively, model-based reinforcement learning offers a solution where dynamics models are trained from historic data and rollouts from the model are then used for policy learning or planning (Deisenroth & Rasmussen, 2011; Chua et al., 2018; Osinski et al., 2020). In the past, machine learning algorithms have been used to directly model plasma dynamics (Char et al., 2023b; Abbate et al., 2021; Boyer

---

[*]Equal contribution  [1]Robotics Institute, Carnegie Mellon University, Pittsburgh, PA, USA [2]Princeton University, Princeton, NJ, USA [3]Princeton Plasma Physics Laboratory, Princeton, NJ, US. Correspondence to: Rohit Sonker <rsonker@andrew.cmu.edu>.

et al., 2021). Reinforcement learning policies have also been trained in models trained solely on fusion data (Char et al., 2023a; Wakatsuki et al., 2023; Degrave et al., 2022). However, the performance of these approaches crucially hinges on the assumption that the data faithfully captures the model at test time. This is problematic in the case of tokamak dynamics, where time-dependent model changes cannot be neglected. Though this issue can be potentially addressed by updating the model with new data, the scarcity of experiments implies that too little data is typically produced to reliably update the model.

In low-dimensional settings, the obstacles posed by conventional RL methods can potentially be addressed by Bayesian optimization (BO). BO is a data-efficient tool for optimizing black box functions (Garnett, 2023). By quantifying model uncertainty, BO achieves a tradeoff between exploration and exploitation, leading to fast convergence in many practical settings (Shalloo et al., 2020; Shields et al., 2021). In the case of tokamak control, BO has been used, e.g., to control the rampdown of a real tokamak (Mehta et al., 2024), and to control neutral beams in a tokamak simulator (Char et al., 2019). However, the work of (Mehta et al., 2024) does not address critical plasma instabilities, whereas (Char et al., 2019) relies on a simulator. Moreover, these methods use a poorly specified prior and require an extensive amount of experiments to perform well.

Motivated by the strengths and shortcomings of existing machine learning-based approaches for tokamak control, we design a novel approach that combines a dynamic model predictor and Bayesian Optimization. Our approach employs a multi-scale approach: a recurrent probabilistic neural network models the high-frequency model dynamics, while a Gaussian process models the effect of low-frequency marginal statistics on the dynamics. After adequate preprocessing, we use historical data to train both models, where the dynamic model serves as a prior for the Gaussian process. Additionally, by leveraging physics-informed assumptions, we design a low-dimensional state space for the Gaussian process. This naturally leads to a contextual Bayesian optimization algorithm tailored to the task at hand, allowing it to find stabilizing actions in a highly data-efficient manner. Moreover, due to its ability to perform fast updates, it allows us to efficiently leverage small batches of data collected during experiments to best inform new decisions on the fly.

We test our approach on a large dataset from past tokamak experiments, where we can quickly identify stable configurations, outperforming a naive approach based exclusively on the recurrent neural network model. Furthermore, we apply our approach in a real-world on the DIII-D Tokamak to find stabilizing actions for a high performing plasma scenario. High performing plasma scenarios need to maintain high temperature and pressures for increased energy, hence, they are more unstable. Our method was able to find stabilizing ECH actuator values in four of eight experiments despite changes to other actuators, a 117% improvement compared to historical experiments with the same configuration.

Our paper is structured as follows: first, we provide some necessary background to nuclear fusion, and define our problem mathematically. Then we discuss our complete pipeline and methodology, followed by the results and analysis on offline historical data and live experiments on a Tokamak reactor. Finally, we provide conclusions and discuss opportunities for future work. Additional details are provided in the Appendix.

## 2. Background and Problem Statement

In this section, we first provide some background on nuclear fusion and then present the formal problem statement. We also include more details in the Appendix section.

### 2.1. Nuclear Fusion

Nuclear fusion is seen as a promising solution for clean, limitless energy, producing no high-level radioactive waste. Among the fusion technologies, tokamaks are the most advanced, using magnetic fields to confine hot plasma to enable fusion conditions. Many countries have invested in tokamak research facilities and currently more 35 nations are collaborating to build ITER, a global project aiming to demonstrate the viability of large-scale commercial fusion reactors (Mohamed et al., 2024; Shimada et al., 2007).

One of the key challenges in tokamak development is plasma disruptions, which can cause severe damage to reactor walls and components, particularly in larger reactors like ITER (Schuller, 1995; Lehnen et al., 2015). These disruptions often stem from tearing mode instabilities (or tearing modes), where magnetic islands form, leading to energy loss and instability. Prior work proposes avoiding tearing instability with predictive models using real-time control (Fu et al., 2020) and reinforcement learning (Seo et al., 2024). However, these methods reduce neutral beam power and add torque to stabilize the plasma. This is undesirable, as reducing beam power leads to lower confinement energy, decreasing the total energy output of the tokamak. On the other hand, adding torque to large tokamaks is itself a challenging issue.

Electron Cyclotron Heating (ECH) has shown promise in counteracting tearing instabilities by driving localized currents at the site of instability (Gantenbein et al., 2000; Kolemen et al., 2014). These and other findings have motivated the inclusion of gyrotrons capable of delivering ECH in future reactors to potentially control tearing instabilities, e.g., ITER will have over 40 gyrotrons. So far, the best results

for stabilizing instabilities with ECH have been achieved by keeping the ECH constant over time, as this minimizes the chance of plasma disruptions. However, how to best deploy ECH is still an open question, as researchers often struggle to find ECH profiles to suit their goals. An ECH profile represents the heating achieved by the gyrotrons across the cross section of the plasma. This can be seen in Fig 4.

In this work, we aim to find feedforward ECH profiles to avoid tearing instability (or modes) in high $q_{min}$ tokamak scenarios. EC waves from gyrotrons lead to EC heating (ECH) profiles and EC current drive (ECCD). The amount of current drive depends on angle of incidence on the plasma. In all experiments, we set angle to produce both ECH and ECCD. More details on effect of ECH on plasma are given in A.1. Pre-emptive suppression of tearing modes with ECCD has been previously shown in (Bardóczi et al., 2023), where ECCD was used at q=2 location. Our goal in this work is to optimize the size and locations of ECH profiles, conditioned on the state of the plasma. During our experiments we focus on the High $q_{min}$ plasma scenario. This is a scenario that supports long duration steady-state plasma operations, making it crucial for future commercial fusion reactors. More details on High $q_{min}$ scenario are provided in the Appendix. We also focus our attention on 2-1 tearing instability, a type which is the most common and significantly disruptive.

## 2.2. Problem Statement

We treat the tokamak dynamics as an unknown discrete-time stochastic system

$$s_{t+1} \sim \Pi_{s_t, a_t}, \tag{1}$$

with states $s_t \in \mathcal{S}$ and actions $a_t \in \mathcal{A}$, and the probability of a tearing mode occurring follows a Bernoulli distribution, parameterized by the tokamak states and actions

$$T_t \sim \text{Bernoulli}(p(s_t, a_t)). \tag{2}$$

Of the state variables describing the plasma, the most important for our approach is the normalized plasma pressure $\beta_{N,t} \in s_t$. A full description of the state space is given in the appendix. The action vector can be decomposed into three different sub-vectors

$$a_t := \left[ a_t^f, \ a_t^c, \ a_t^g \right] \tag{3}$$

as follows. The actions $a_t^f$ correspond to feedforward inputs specified before the experiment. These correspond, e.g., to gas flows, plasma density, and shape controls. They are typically picked manually based on the success of previous experiments. The actions $a_t^c$ are part of a feedback control loop that aims to stabilize the normalized plasma pressure $\beta_{N,t} \in s_t$, arguably one of the most important quantities since it measures the efficiency of plasma confinement relative to the magnetic field strength. The third set of actions

$a_t^g$ corresponds to gyrotron angles, operated at constant power, which we use to keep the tearing instability from occurring. The gyrotrons operate on the plasma by generating an ECH profile $a_t^{\text{ech}} = \phi(a_t^g)$. Unlike $a_t^f$ and $a_t^c$, the number of gyrotrons, i.e., the dimension of $a_t^g$, potentially changes between each individual experiment. This is due to various reasons, e.g., due to hardware issues or because some gyrotrons might be required for other tasks, such as elm suppression or density control (Hu et al., 2024; Ono et al., 2024).

This paper considers the case where the gyrotron angles $a_t^g$ are kept fixed throughout each experiment rollout, i.e., $a_0^g = a_1^g = ... = a_\tau^g =: a^g$, where $\tau$ is the length of the rollout horizon. This is a common operating mode and also a design choice, which we make because we need to search as efficiently as possible within the action space, an impossible task if its dimension is too large. Thus, the feedforward actions $a_t^f$ and the target normalized plasma pressure $\bar{\beta}_N$, which defines the set-point for $a_t^c$, are specified beforehand and can change between rollouts. Our goal, is to then select $a^g$ separately for each experiment such that the probability of encountering a tearing mode $T_t = 1$ is minimized over the full rollout horizon.

## 3. Methodology

We now introduce our method, DynaBO, which aims to find stationary feedforward ECH profiles that mitigate tearing instabilities. Our complete pipeline is illustrated in Fig. 1. On a high level, the process is as follows - We model the system at two different time scales to inform the choice of actuator commands for each experiment. At a smaller, more granular time scale, we use a recurrent probabilistic neural network model (RPNN) to estimate the high-frequency behavior during each experiment. The coarser model corresponds to a Gaussian process model and is trained to predict the behavior of the system based on marginal statistics from experimental observations and RPNN predictions of the objective function, which act as a prior mean. In this case, the objective function is the time-to-tearing instability. Given the target normalized plasma pressure $\bar{\beta}_N$, we leverage the Gaussian process to select actions (ECH profiles) in a low-dimensional space via Bayesian optimization. The desired profile is then converted to gyrotron angles and applied to the tokamak. Finally, we update our model with the resulting time-to-tearing instability and actual ECH profile and repeat the procedure. We update the model with the measured ECH profile because it can diverge significantly from the desired one. In the following sections, we discuss the individual components of our method - the high-frequency RPNN and instability predictor, the Gaussian Process model, and action selection by Bayesian optimization.

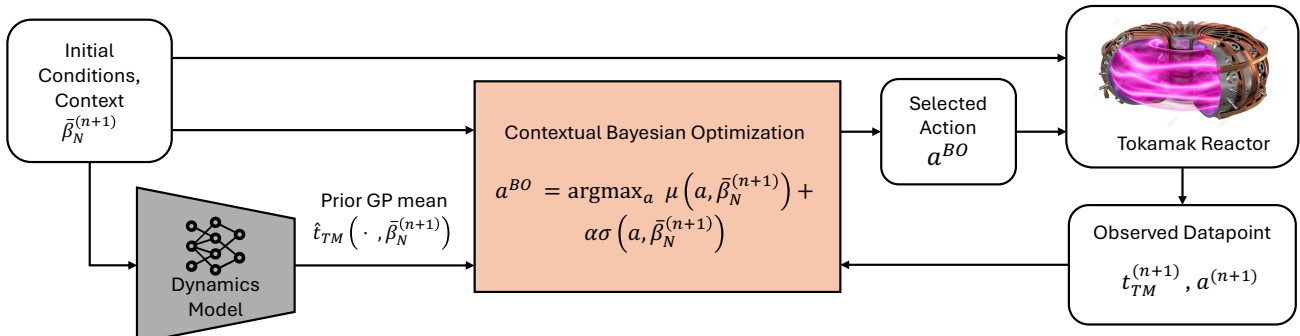

*Figure 1.* DynaBO Pipeline to generate feedforward trajectory actions. Initial conditions and feedforward actuators are first used by the RPNN to generate rollouts through which we compute the prior mean of the objective function (time to tearing instability). Our Bayesian optimization algorithm uses this to optimize for actions (ECH). Noisy outputs from the Tokamak are then used to update the Gaussian process model used for Bayesian optimization.

### 3.1. Recurrent Probabilistic Neural Networks and Instability Prediction

We employ a Recurrent Probabilistic Neural Network (RPNN) (Char et al., 2023b) to model the high-frequency behavior of the tokamak. An RPNN has a Gated Rectifier Unit (GRU) cell, which stores information about past states and actions. We use this network as a state transition model which takes current state and action and outputs a probability distribution over the next state. Given $s_t$ and $a_t$ as inputs, the RPNN outputs a multivariate normal distribution with mean $\eta$ and variance $\Sigma^2$, which we employ to approximate the system dynamics:

$$\mathcal{N}(\eta(s_t, a_t), \Sigma^2(s_t, a_t)) \approx \Pi_{s_t, a_t}. \tag{4}$$

To bypass the issue that the number of gyrotrons differs for each rollout in the training dataset, we assume that the resulting heating profiles $a_t^{\text{ech}}$ can be controlled directly, allowing us to disregard $a_t^g$ both in training and testing. Moreover, to facilitate optimization, we approximate the ECH profile with a stationary Gaussian curve $a_q$, yielding the approximate ECH profile $a_q \approx a_t^{\text{ech}}$ for all $t$. When carrying out experiments, we then project $a_q$ onto $a_t^g$, which can be done for an arbitrary number of gyrotrons, i.e., for an arbitrary dimension of $a_t^g$.

In addition to the RPNN, we train a binary classifier, which we call the tearing mode predictor $h$ to predict the probability of a tearing mode occurring based on input state and actuator.

$$h(s_t, a_t) \approx \text{Bernoulli}(p(s_t, a_t)). \tag{5}$$

### 3.2. Gaussian Process Model

Using RPNN rollouts exclusively for experimental design is challenging for various reasons. Although the RPNN accurately captures some of the tokamak behavior, the resulting predictions often exhibit significant errors, largely due to the sim2real gap caused by time-dependent fluctuations in the environment variables, e.g., due to maintenance or hardware changes provoked by previous experiments. Furthermore, retraining the RPNN between experiments and using it to select ECH profiles $a_q$ is virtually impossible because the newly collected data is too small and we only have a few minutes between experiments.

We address the above-mentioned issues by employing a Gaussian process (GP) model, a nonparametric model that is very data-efficient, especially in low-dimensional spaces (Deisenroth & Rasmussen, 2011). A GP corresponds to an infinite collection of random variables, of which any finite number is jointly normally distributed. To fully leverage the strengths of GP models, we need to carefully summarize the information collected between experiments before training the GP. This is done as follows.

First, we assume the achieved normalized plasma pressure $\beta_N$ is independent of the ECH profile $a_q$. This is a reasonable assumption because $\beta_N$ is largely determined by neutral beams, which are controlled through the feedback variables $a_t^c$. We then approximate the feedforward and feedback control actions $a_1^f, \ldots, a_\tau^f$ and $a_1^c, \ldots, a_\tau^c$ by assuming that they are uniquely specified by the target normalized plasma pressure, denoted by $\bar{\beta}_N$. This choice is partly justified because the feedforward and feedback control actions are often primarily informed by a target normalized plasma pressure. Finally, we employ the GP to predict the time-to-tearing mode $t_{\text{TM}}$, which we use as a proxy for the probability of a tearing mode occurring. The rationale behind this choice is twofold. First, a scenario where a tearing instability occurs late implies a higher degree of stability than a scenario where it occurs earlier. Moreover, this allows us to use the GP in a regression setting, where GPs are strongest and best understood. The GP inputs are thus $\bar{\beta}_N$ and $a_q$, whereas the output is $t_{\text{TM}}$.

The GP is fully specified by a prior mean function $m$ and a kernel $k$ that specifies the similarity between training inputs. In this work, we employ a squared-exponential kernel $k$, which is appropriate for approximating most continuous functions. The prior mean $m$ corresponds then to the average time-to-tearing mode $\hat{t}_{\mathsf{TM}}$ predicted by autoregressive rollouts of the RPNN and tearing mode predictor:

$$\hat{t}_{\mathsf{TM}}(\bar{\beta}_N, a_q) := \mathbb{E}\left( \arg\min_t t \,\middle|\, T_t \geq 0.5,\ T_t \sim h(s_t, a_t), \right.$$

$$\left. s_{t+1} \sim \mathcal{N}\big(\eta(s_t, a_t), \Sigma^2(s_t, a_t)\big),\ \bar{\beta}_N = \frac{1}{\tau}\sum_{t=1}^{\tau} \beta_{N,t} \right)$$

The dataset used to train the GP has the form

$$\mathcal{D}_n = \{\bar{\beta}_N^{(i)}, a_q^{(i)}, t_{\mathsf{TM}}^{(i)}\}_{i=1,\dots,n}.$$

We use it to compute the posterior distribution of $t_{\mathsf{TM}}$ for arbitrary test inputs $\bar{\beta}_N^*, a_q^*$, which corresponds to a normal distribution with mean and covariance

$$\mu_n(\bar{\beta}_N^*, a_q^*) = \hat{t}_{\mathsf{TM}}(\bar{\beta}_N^*, a_q^*) + k_*^\top (K + \sigma^2 I)^{-1} \Delta_n, \quad (6)$$

$$\sigma_n^2(\bar{\beta}_N^*, a_q^*) = k_{**} - k_*^\top (K + \sigma_{\mathrm{no}}^2 I)^{-1} k_* + \sigma_{\mathrm{no}}^2, \quad (7)$$

respectively. Here $\sigma_{\mathrm{no}}^2$ is the noise variance, $[k_*]_i = k(\bar{\beta}_N^*, a_q^*, \bar{\beta}_N^{(i)}, a_q^{(i)})$, $[K]_{ij} = k(\bar{\beta}_N^{(i)}, a_q^{(i)}, \bar{\beta}_N^{(j)}, a_q^{(j)})$, $k_{**} = k(\bar{\beta}_N^*, a_q^*, \bar{\beta}_N^*, a_q^*)$. The vector $[\Delta_n]_i = t_{\mathsf{TM}}^{(i)} - \hat{t}_{\mathsf{TM}}(\bar{\beta}_N^{(i)}, a_q^{(i)})$ contains the difference between the observed and the predicted time-to-tearing mode. In practice, the posterior variance $\sigma_n^2$ is typically small when evaluated in distribution and larger when out of distribution. Hence, intuitively, the posterior GP mean $\mu_n$ can be viewed as the predictive model, whereas $\sigma_n^2$ quantifies model uncertainty. This distinction is important for understanding Bayesian optimization, which we introduce in the next section.

### 3.3. Contextual Bayesian Optimization with Noisy Inputs

Contextual Bayesian optimization is a data-efficient tool that leverages GPs to optimize black-box functions. Given a context that specifies the environment, it optimizes an acquisition function that carefully balances exploration versus exploitation. By recursively updating the acquisition function after every observation, it gradually becomes more confident about its predictions, resulting in convergence. In every experiment, we treat the target normalized plasma pressure $\bar{\beta}_N^{(n+1))}$, specified before the experiment, as the context and choose the ECH profile by optimizing the so-called upper confidence bound (UCB) acquisition function

$$a_q^{\mathsf{BO}} = \arg\max_{a_q} \mu_n(a_q, \bar{\beta}_N^{(n+1))}) + \alpha\sigma_n(a_q, \bar{\beta}_N^{(n+1)}), \quad (8)$$

where $\alpha$ balances exploration and exploitation. In conventional BO methods, the next step consists of setting $a_q^{(n+1)} = a_q^{\mathsf{BO}}$, measuring the time-to-tearing mode $t_{\mathsf{TM}}^{(i+1)}$, and updating the GP accordingly. However, in our setting there is the added challenge that the target plasma $\bar{\beta}_N$ and the desired ECH profile corresponding to $a_q^{\mathsf{BO}}$ are not reproduced exactly. This is due to various reasons, including a changing number of available gyrotrons, hardware failures, actuator noise, and unmodeled disturbances. To alleviate this issue, we measure the ECH profile and $\bar{\beta}_N$ obtained during the experiment and treat them as the true inputs $a_q^{(n+1)}$ and $\bar{\beta}_N^{(n+1)}$ used to update the GP model. Formally, this is equivalent to standard contextual BO where the GP inputs $a_q$ and context $\bar{\beta}_N$ in equation 8 are perturbed by unknown noise.

## 4. Results

This section presents results from offline tests using historical data and results from experiments at the General Atomics DIII-D Tokamak Fusion Facility. We use a fixed RPNN in all experiments, trained using $15,000$ one-step state transition observations collected between 2010 and 2019 at the DIII-D tokamak. More details on the dataset can be found in A.2.

Through our analysis of offline and online experiments, we aim to answer the following questions:

1. How does DynaBO compare to other baselines? How robust is it in terms of the choice of kernel?

2. Can DynaBO find heating profiles that avoid tearing instabilities altogether using only a handful of trials? If not, can it prolong the stable operation time of the plasma?

We address question 1 by conducting simulated experiments from offline data and comparing performance across all methods. We then address question 2 with results from live experiments on the DIII-D Tokamak. As we show in the following, both questions have an affirmative answer.

### 4.1. Offline Data Analysis

This section employs historical data from the DIII-D tokamak to compare DynaBO with several baselines. Specifically, we employ data from 281 past experiments carried out at the DIII-D tokamak between 2012 and 2023. We selected the data points based on their similarity with our live experiment, particularly the range of $\bar{\beta}_N$ and the high $q_{min}$ specification. Appendix A.2.2 provides a detailed description of the selection procedure.

We employ the historical data to emulate our live experiment from Section 4.2. This is achieved as follows. At every

time step, we sample the target plasma pressure $\bar{\beta}_N$ from a uniform distribution corresponding to the range of the historical data and condition DynaBO on $\bar{\beta}_N$. We then select a subset from the historical data with plasma pressure values within the interval $[\bar{\beta}_N - \epsilon, \bar{\beta}_N + \epsilon]$, where $\epsilon = 0.04$, and use DynaBO to select ECH profile values corresponding to an element of that subset. After selecting an ECH profile, we treat the historical data point corresponding to that particular ECH profile as a new observation, which we use to update our GP model.

We compare our approach to four different baselines and ablations: the setting where we fully trust the RPNN to predict tearing modes without updating it, a vanilla GP with a zero-mean prior, a vanilla GP with a prior value which represents mean of the past data and our approach using a time-dependent kernel. The motivation for the latter approach is that one could naively trust older data less than that closer to present-day experiments, attempting to model changes made due to deliberate hardware changes and repairs. In addition, we consider a linear kernel, Gaussian kernels, and Matérn kernels with different hyperparameter configurations to analyze our approach's robustness under varying model specifications.

In Fig.2, we depict the cumulative regret

$$\text{Cumulative Regret}(N) = \sum_{i=1}^{N} (\tau^{\text{max}} - t_{\text{TM}}^{(i)}), \qquad (9)$$

where $\tau^{\text{max}} = 10s$ is the maximal shot length. As can be seen, DynaBO and DynaBO with time dependency achieve the highest performance in all settings except the linear kernel setting. By contrast, the RPNN-based method and vanilla BO (with both zero and mean value priors) cannot consistently find good solutions despite performing more steps than the total number of data. This indicates that DynaBO does not become overconfident and is robust to the choice of kernel and hyperparameters except when the kernel is clearly misspecified, e.g., when using a linear kernel. While including time as an input to the GP performs competitively, overall, the improvement seems only marginal. One possible explanation is that the reliability of the data depends on multiple factors, many of which cannot be explained exclusively as a continuous function of time, e.g., sensor and actuator upgrades, particle absorption and release by the tokamak wall, and the presence of impurities which may have been used in preceding experiments.

We note that the vanilla GP does converge after more than 500 steps, i.e., after the plots in Fig. 2 end. However, such a long convergence time is unacceptable for our setting since fusion experiments are very costly, and we only get a handful of experiments to explore.

To compare exploration, in Fig. 3, we display the ECH profiles queried by the different baselines using a Gaussian

kernel. DynaBO and DynaBO+time exhibit more variety in the queried ECH profiles than in the vanilla GP and the RPNN baseline. This corresponds to better exploration, resulting in lower regret for our approach. We observed similar trends using all other kernels except the linear one.

We additionally tested different acquisitions functions for our problem setup A.4. However, we finally chose Upper Confidence Bound (UCB) as our acquisition function because it allows easy tuning of exploration vs exploitation during the actual experiment.

### 4.2. DIII-D Tokamak Experiments

We tested our algorithm at the DIII-D Tokamak hosted by General Atomics during a two-hour time window allocated to us during the FY24 campaign. Each experiment run at DIII-D is known as a 'shot'. Each shot is then assigned a unique shot number.

To make the most of our time and make significant statements about results, we opted for a pre-specified set of feedforward actuators $a_t^f$ that represents high performance experiments in high $q_{min}$ scenario.

In addition to our dynamics model which is trained on larger set of historical data, we further conditioned our GP on 125 historic high $q_{min}$ experiments. Appendix A.2.2 provides an overview of the data used to train the GP. Our experiment consisted of 8 BO iterations with DynaBO.

After each run of DynaBO, the selected heating profiles $a_q^{\text{BO}}$ were converted to gyrotron angles and entered into the Plasma Control System, the interface that controls the tokamak. After a few seconds of maintaining the plasma, we ramped down the actuators and terminated the shot.

We started our experiments by recreating a high-performing historical high $q_{min}$ experiment with a tearing instability. For this, we recreated the conditions in shot 180636, a plasma shot executed previously at DIII-D. Once this shot with positive tearing instability was recreated, we ran additional shots where we varied the ECH using DynaBO while keeping the remaining settings identical. The details for each of the 8 shots carried out using DynaBO are shown in Table 1. As can be seen, our algorithm was able to avoid tearing instabilities in 4 out of 8 shots successfully. Moreover, we maintained a stable plasma for 3339 milliseconds on average. Although this number of shots is too low to be statistically significant, we stress that the chosen configuration is very challenging. For reference, there were 61 historical experiments at DIII-D with very similar settings (high $q_{min}$ experiments that maintain high normalized plasma pressure $\beta_N > 3.0$). Of those experiments, 47 reported tearing instabilities, corresponding to a tearing instability rate of 77%. This set of experiments is selected as a baseline considering that these experiments also use ECH with current drive (sim-

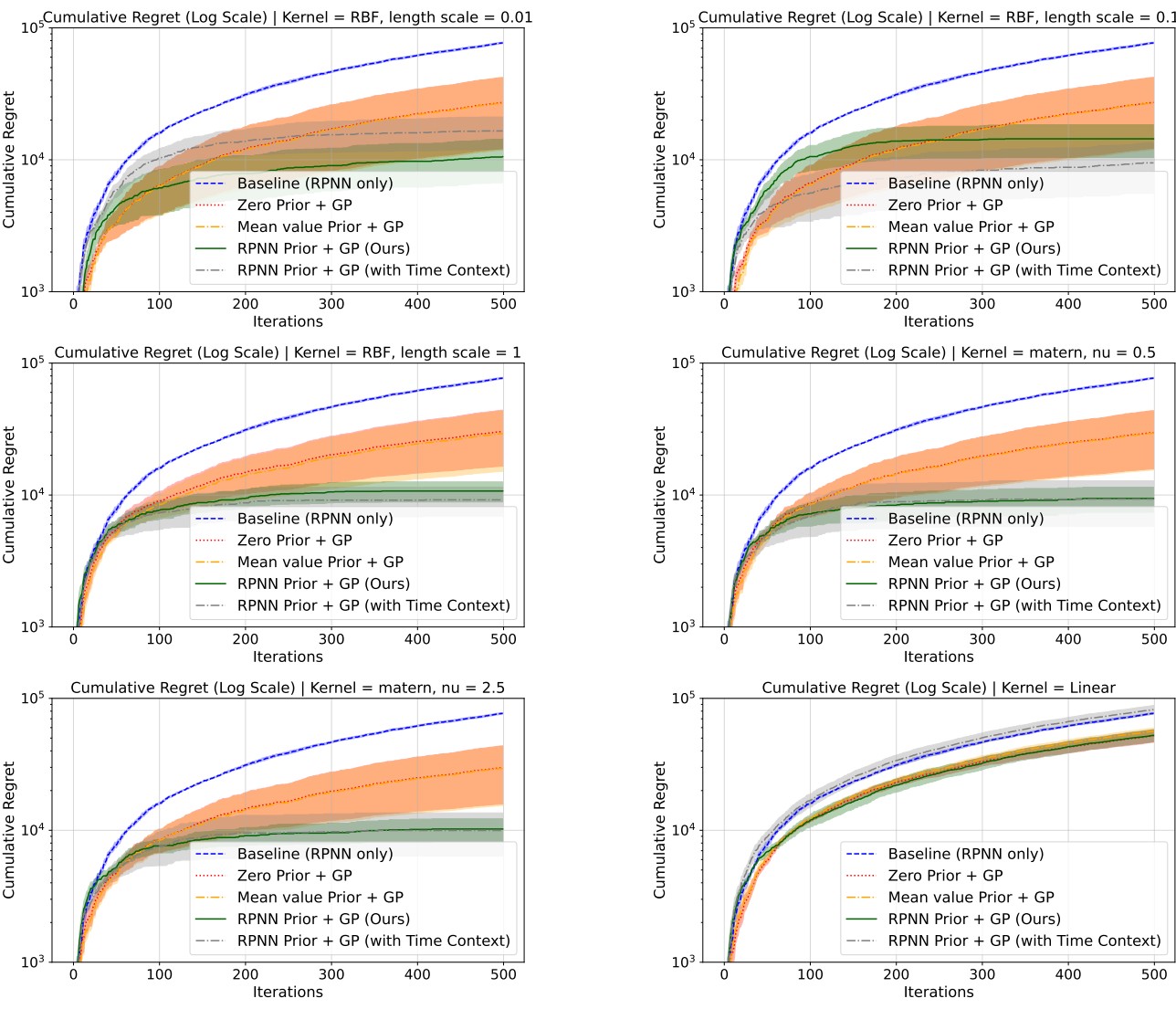

*Figure 2.* Cumulative Regret (log scale) achieved by DynaBO (green), DynaBO with a time-dependent kernel (gray), the RPNN only (blue), vanilla GP with a zero-mean prior (red) and vanilla GP with mean value as prior (orange) using six different kernels.

ilar to our experiment), however, they use manual methods for selecting ECH profiles. The average time to tearing instability in these experiments is 2424 milliseconds, well below our average of 3339 milliseconds. More details on identification of tearing mode instabilities in our experiments from raw signals is shown in Section A.5.

## 5. Limitations and Future Work

While our method is shown to preemptively suppress tearing instabilities, it is mainly data-driven, and potential improvements are possible by incorporating physics knowledge. One possible solution is to develop physics-informed neural network models, such as incorporating elements of the Rutherford equation to improve interpretability. Another

shortcoming is that our current method is only applicable to feedforward control scenarios. This means the algorithm cannot adapt to unexpected real-time changes in the plasma, such as MHD activity or impurity changes. In future work, we aim to extend our learning to feedback control systems.

## 6. Conclusion

In this work, motivated by the challenges of tokamak control, we develop a multi-scale modeling approach for decision making on complex real world systems with limited availability of data. Our pipeline leverages a high-frequency neural network model of the system dynamics and a Gaussian process that makes predictions based on marginal statistics. Together, both models form a Bayesian optimization

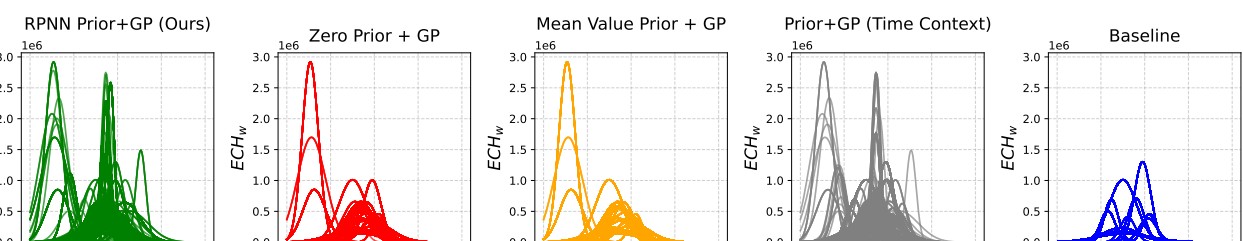

*Figure 3.* ECH Profiles queried by different methods during simulated offline runs using a Gaussian kernel. We see that DynaBO and DynaBO with a time-dependent GP explore the most, highlighting the importance of our dynamic model prior mean.

| Experiment ID (Shotnumber) | Target $\bar{\beta}_N$ | Tearing Instability Avoided | Stability Time (ms) |
|---|---|---|---|
| 199599 | 3.37 | **Yes** | 4566 |
| 199601 | 3.27 | **Yes** | 4632 |
| 199602 | 3.27 | No | 2107 |
| 199603 | 3.27 | No | 2149 |
| 199604 | 3.27 | **Yes** | 4592 |
| 199605 | 3.14 | No | 1512 |
| 199606 | 3.45 | No | 3512 |
| 199607 | 3.43 | **Yes** | 3654 |

*Table 1.* Results from two-hour experiments at DIII-D Tokamak : DynaBO avoids tearing modes in 4/8 runs in a high-performance configuration with a historical rate of occurrence of the tearing instabilities of 77%. The mean time under stability with DynaBO is 3339 ms while the historical time under stability is 2424 ms corresponding to a 914 ms improvement in stability duration. Generally, for stable experiments at DIII-D, the plasma stability is maintained for 4-5s.

algorithm tailored to the task at hand that can quickly identify stabilizing control actions. This is achieved by making decisions on the fly based on newly collected data. On a historical data set, our method outperforms vanilla BO and a naive baseline that relies exclusively on neural network predictions. This is mainly due to our approach having better exploration capability. Moreover, our method shows promise in live experiments on the DIII-D Fusion reactor. During the experiments, our approach successfully avoided tearing instability in 4/8 runs despite highly unstable conditions, representing an improvement of over 117% compared to past experiments.

## Acknowledgments

We would like to thank the exceptional staff at the DIII-D National Fusion Facility that helped with the preparation and running of this experiment. This material is based upon work supported by the U.S. Department of Energy, Office of Science, Office of Fusion Energy Sciences, using the DIII-D National Fusion Facility, a DOE Office of Science user facility, under Awards DE-FC02-04ER54698, DE-SC0024544 and DE-SC0015480.

## Disclaimer

## Impact Statement

Our work illustrates the potential of combining complex high-frequency and low-frequency models to address challenges of complex real world systems like Tokamaks. Solving nuclear fusion is one of the most important goals of our time and our work aims to utilize machine learning research methods to advance this field. The need for such methods in this area will increase in the future, as new and larger reactors such as ITER become operational, and a significant gap between existing and new models needs to be bridged with very little data. This is the case not only for the stabilization setting considered in this paper but also for settings such as ramp-up design, where a different set of actuators is considered. Moreover, we believe this approach could be of interest to several other applications where the discrepancy between past and present data is considerable, e.g., complex physical systems with changing dynamics.

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

# A. Appendix

## A.1. Additional Fusion Background

*Effect of ECH on Plasma :* Electron cyclotron heating waves released by gyrotrons interact with the plasma by being absorbed by electrons whose gyrofrequency matches the frequency of the ECH wave. This absorption is highly localized, making electron cyclotron heating (ECH) a powerful tool for precise plasma control. ECH increases the electron temperature at the absorption point and often reduces electron density, a phenomenon known as density pump-out (Wang et al., 2017). As density decreases, plasma rotation tends to speed up due to reduced inertia. By adjusting the toroidal injection angle of the EC wave, different effects can be achieved. When injected perpendicular to the toroidal direction, EC waves primarily heat the plasma. When injected parallel, they drive electron acceleration in the direction of the plasma current, a process called electron cyclotron current drive (ECCD). While the injection angle can be switched between shots, it cannot be actively adjusted during a single shot. In this experiment, the toriodal angle is set, such that we get both ECH and ECCD profiles which are of similar shape.

*Details on high $q_{min}$ plasma scenario :* The high $q_{min}$ scenario refers to a group of scenarios related with elevated values of $q_{min}$, the minimum value of the safety factor profile. Under this umbrella of scenarios, there are three main groups: $q_{min} = 1.4$, $q_{min} = 1.5 - 2$, and $q_{min} > 2$. The lowest of the range with $q_{min} = 1.4$ has shown promise as being stable to TMs, but did not have the greatest confinement, while the highest of $q_{min} > 2$ was stable to TMs but had lower energy confinement. The middle range of $q_{min} = 1.5 - 2$ has very desirable confinement but is very susceptible to TMs (Holcomb et al., 2014). The purpose of this experiment is to work in that middle range of $q_{min}$, referred to as the elevated $q_{min}$ scenario, to stabilize TMs and achieve higher confinement than either of the other similar scenario options. In the elevated $q_{min}$ scenario, 2/1 TMs are the most prevalent mode because they require the least energy to perform magnetic reconnection and form a magnetic island. Other lower order modes like 3/1 or 5/2 can sometimes occur but are significantly less frequent as they require more energy to form a magnetic island. Stabilizing TMs in the elevated $q_{min}$ scenario would show a path forward for this high-confinement scenario as a possible operating scenario for a fusion power plant.

## A.2. Dataset

Plasma trajectories on a Tokamak consists of three phases. The ramp-up phase, where the gases are heated and pressure is increased to generate the plasma state where fusion occurs. During this phase, the normalized plasma pressure $\beta_N$ rises. Then, we enter the flat-top phase, where the plasma pressure $\beta_N$ is sustained, allowing fusion to occur. In this phase, $\beta_N$ is mostly constant and the aim to maintain this state without instabilities. Finally, the actuators are gradually ramped down and the plasma is safely terminated as the shot concludes. In this paper, we stay in the flat top phase and aim to stabilize it. To create our dataset, we hence use only flat top data and only control actuators during this phase of the experiment.

Our complete dataset consists of $\sim 15000$ plasma trajectories from historical experiments at DIII-D Tokamak. The data contains signals from different diagnostics have different dimensions and spatial resolutions, and the availability and target positions of each channel vary depending on the discharge condition. Therefore, the measured signals are preprocessed into structured data of the same dimension and spatial resolution using the profile reconstruction and equilibrium fitting (EFIT). These shots contain many different signals, some of which are described below. The dataset consists of scalar signals defined at every timestep and profile signals which are defined along 33 or 200 points along the radius of the plasma cross-section. These consist of temperature, ion temperature, pressure, rotation, safety (Q) factor and density. For these signals we first convert them into PCA components and select the top components which are able to explain 99% of the variance in data. The Electron Cyclotron Heating (ECH) profile we choose to control, is also defined at 200 points along the plasma radius. PCA is unable to describe ECH profiles, however they can be described well by a Gaussian curve and are hence parameterized by the center, width and amplitude of the curve. These 3 parameters form our parameterization $a_q$ of the ECH profiles. The model state space $s_t$ is shown in table 2 while the actuator space $a_t$ is shown in table 3.

### A.2.1. DATASET FOR DYNAMICS MODEL

For training the RPNN, we utilize this data set with data points every 20 ms in time intervals with trajectories having an average length of 5 seconds. The RPNN is trained to predict $\Delta s_{t+1}$ given $(s_t, a_t)$. We add tearing mode labels to this dataset and train a random forest classifier to predict the probability of tearing modes at every time step. We tried incorporating tearing mode predictions inside the RPNN; however, this did not yield good results.. This is likely due to the formation of spurious correlations and causality issues formed by introducing tearing modes into the dataset.

| State Variables | Dimensions |
|---|---|
| Normalized Plasma Pressure $\beta_N$ | Scalar |
| Line averaged density | Scalar |
| Loop voltage | Scalar |
| Confinement Energy | Scalar |
| Temperature Profile | Decomposed to 4 PCA components |
| Ion Temperature Profile | Decomposed to 4 PCA components |
| Density Profile | Decomposed to 4 PCA components |
| Rotation Profile | Decomposed to 4 PCA Components |
| Pressure Profile | Decomposed to 2 PCA components |
| q Profile (safety factor) | Decomposed to 2 PCA components |

*Table 2.* Plasma Features used as state space for RPNN model.

| Actuator Variables | Dimensions |
|---|---|
| Power Injected | Scalar |
| Torque Injected | Scalar |
| Target Current | Scalar |
| Target Density | Scalar |
| Magnetic Field | Scalar |
| Gas Puffing | Scalar |
| Shape Controls | 6 Scalars |
| ECH Profile | Decomposed to Gaussian curve with mean, stddev, amplitude $(\mu, \sigma, w)$ |

*Table 3.* Plasma Features used as actuator space of the RPNN model.

### A.2.2. DATASET FOR GAUSSIAN PROCESS

To create the dataset for offline testing $\mathcal{D}^H$, we first limit ourselves to High $q_{min}$ trajectories which achieve a high normalized plasma pressure $\beta_N > 3.0$. This constraint follows our experiment conditions. This leaves us with 281 trajectories. We subsequently convert this data from a time-step scale to a trajectory level scale. We take average $\beta_N$ of the flat-top phase of the trajectory. For ECH profile $a_q$, we take a mean of all profiles in the flat-top phase of the experiment. This is the phase where the high-energy plasma state is maintained. We thus get the dataset $D^H$ where $D_i^H$ consists of triplet $(\beta_N^{\ i}, a_q^i, t_{\text{TM}}^i)$ i.e. the normalized plasma pressure, parameterized ECH profile and the observed time-to-tearing mode. This dataset is used for offline testing.

For online testing, we subset this dataset further by only keeping whose ECH profiles are lie in the achievable parameter space as per experiment configuration, which leaves us with 125 training points. This is used as a training set for the Gaussian Process before testing on the real tokamak.

### A.3. Training Details for Recurrent Probabilistic Dynamics Model (RPNN)

**Network Architecture:**

- **Encoder:**
    - Fully Connected (FC) layer: `input_dim` $\times$ 512
    - FC layer: $512 \times 512$

- **Memory Unit:**
    - Gated Recurrent Unit (GRU) block: $512 \times 256$

- **Decoder (with residual connections between FC layers):**
    - FC layer: $256 \times 512$
    - FC layers: $512 \times 512$ (repeated 8 times)

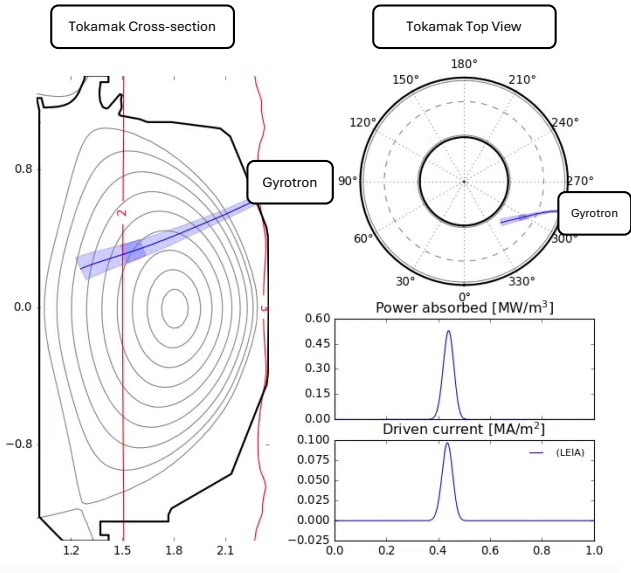

*Figure 4.* Gyrotron action on the Plasma inside the Tokamak. The bottom 2 curves indicate the power absorbed (heating profile) and current driven in the plasma from the centre to outer region of the plasma.

  - FC layer: $512 \times 128$

- **Output Heads:**
  - Mean head: $128 \times$ `output_dim`
  - Log-variance head: $128 \times$ `output_dim`

The network predicts the parameters of a probability distribution, and is trained using a log-likelihood loss. We use the Adam optimizer with a learning rate of $3 \times 10^{-4}$ and a weight decay of $1 \times 10^{-3}$. Early stopping is applied with a patience of 250 epochs based on performance on a validation set comprising 10% of the total data.

### A.4. Comparing Different Acquisition Functions for Offline Experiments

We tested different acquisitions functions for this problem setup. We tested Expected Improvement, Thompson Sampling, and Upper Confidence Bound (UCB). There results of running the offline experiments are shown below. All acquisitions functions gave similar order cumulative regret at the end of 500 epochs. We finally chose UCB as our acquisition function because it allows easy tuning of exploration vs exploitation during the actual experiment.

| Acq. Fun. | RBF (ls = 0.1) | | RBF (ls = 1) | | Matern ($\nu = 2.5$) | |
|---|---|---|---|---|---|---|
| | Mean | STD | Mean | STD | Mean | STD |
| UCB | 14418.2 | 8038.94 | 10726.4 | 3717.18 | 10223.0 | 3997.86 |
| Thompson Sampling | 11184.0 | 2835.95 | 15696.6 | 4091.07 | 18139.6 | 5818.07 |
| EI | 8201.0 | 10183.31 | 10342.6 | 9859.99 | 10231.8 | 9582.02 |

*Table 4.* Cumulative regret values at the end of 500 epochs under offline testing setup for different acquisition functions across different kernels and length scales. The orders of cumulative regret are similar across all acquisitions functions.

### A.5. Details about Online Experiment Results

In this section, we analyze the signals from our experiment runs at DIII-D. The results are shown in Fig. 5 and Fig. 6. We show the n1rms signal which measure the n=1 magnetic perturbations. We also show the normalized plasma pressure $\beta_N$, a quantity which directly corresponds to energy confinement levels in the plasma. For experiments 199606-199607, it is tricky

to detect a tearing instability, hence we also add power injected. A sustained high value in n1rms along with drops in $\beta_N$ usually denotes tearing modes. However, $\beta_N$ drops may vary depending on severity.

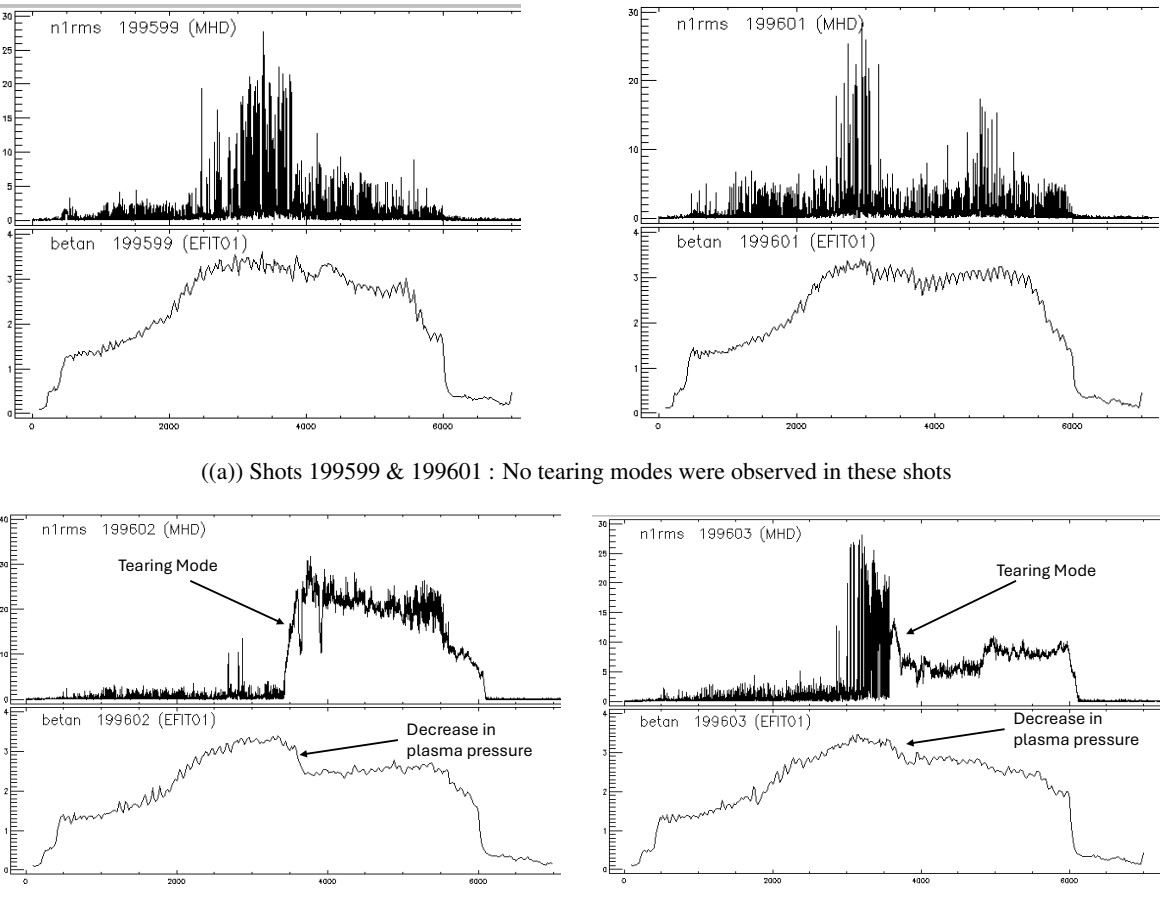

((a)) Shots 199599 & 199601 : No tearing modes were observed in these shots

((b)) Clear Tearing mode happens in 199602 which leads to loss in normalized plasma pressure $\beta_N$. In 199603 we see a tearing mode form however its difficult to spot as the loss in $\beta_N$ is minor.

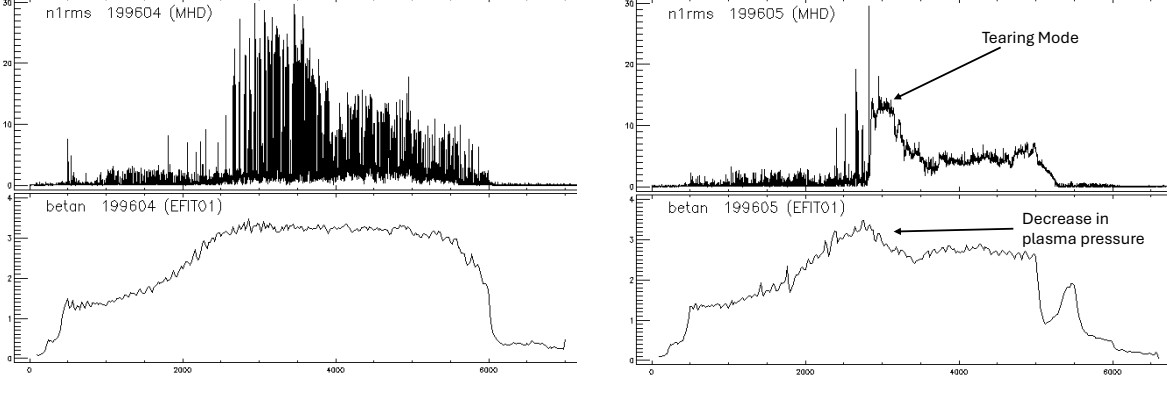

((c)) No tearing mode happens in 199604. A tearing mode happens in 199605

*Figure 5.* Identifying Tearing modes from Raw signal data. We use n1rms signal (denotes magnetic perturbations) and normalized plasma pressure $\beta_N$ to identify tearing modes. A sustained high n1rms signal denotes tearing modes. We label the drop in $\beta_N$ due to tearing mode formation. Note that in all experiments, $\beta_N$ drops towards then end as power injected is dropped to safely end the experiment

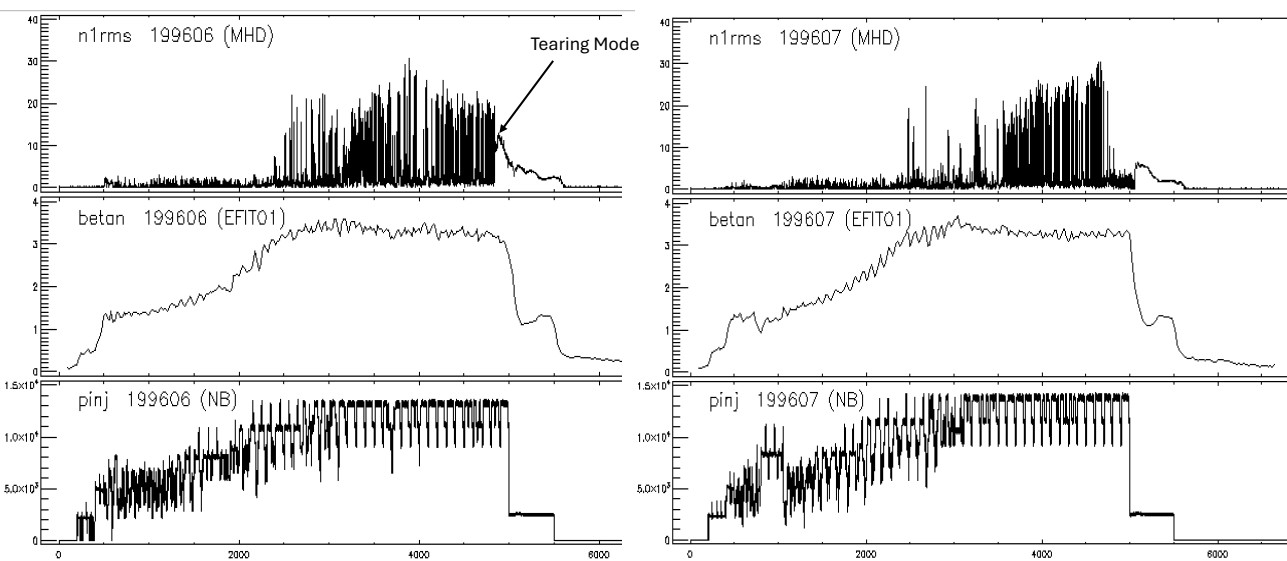

*Figure 6.* In this figure we also include the power injected (pinj) along with n1rms and $\beta_N$. in 199606, we see a very late tearing mode which occurs just before power injected is dropped. Very low loss in $\beta_N$ is seen due to the tearing mode. Finally, in 199607 no tearing modes are seen.

### A.6. Approximating the Prior

The historical data used to train the RPNN and the GP does not contain the target normalized plasma pressure $\bar{\beta}_N$. Instead, it only contains the actions $a_t$ achieved during the shot. Similarly, the RPNN is trained exclusively on the actions, and not on $\bar{\beta}_N$, hence a direct mapping from $\bar{\beta}_N$ does not take place in the RPNN. In the experiments, we address these issues as follows. In the historical data, we set $\bar{\beta}_N$ to be equal to the average normalized plasma pressure, i.e.,

$$\bar{\beta}_N^{(i)} \approx \sum_{t=1}^{\tau} \beta_{N,t}{}^{(i)}. \tag{10}$$

This is a reasonable assumption since the target $\bar{\beta}_N$ is mostly achieved in practice. We then approximate the time-to-tearing mode $\hat{t}_{\text{TM}}(\bar{\beta}_N, a_q)$ predicted by the RPNN given $\bar{\beta}_N$ and $a_q$ as follows. We first use $a_q$ to compute the actions $a_t^{\text{ech}}$. We then compute the remaining actions $a_t^c$ and $a_t^f$ by sampling full rollouts from the historical data and setting $a_t^c$ and $a_t^f$ equal to the corresponding actions. We then look at the resulting average normalized plasma pressure and set it equal to $\bar{\beta}_N^{(i)}$. We do this for all ECH actions $a_q$ within a $10 \times 10 \times 10$ grid within the space of ECH parameters, specified by the historically largest and smallest parameter values in the historical data set. We then separate the results into bins that have the same value of $\bar{\beta}_N^{(i)}$ up to a margin of $\epsilon = 0.04$, and average over all tearing modes within that bin, yielding $\hat{t}_{\text{TM}}(\bar{\beta}_N, a_q)$. At test time, we project all points to the closest point on the grid, both when performing queries and before updating the GP model.

### A.7. Conversion of ECH Profile to Gyrotron Angles

Even though we selected ECH profiles as our action space, the Plasma Control System (PCS) at DIID tokamak expects the output to be Gyrotron angles, which denote locations where they will be aimed. To make this conversion, we used OMFIT software (Meneghini et al., 2015). We selected ECH profiles as our action space instead of gyrotron angles because at experiment time one does not know how many gyrotrons are available. With this choice of action space, we ensure our method is agnostic of number of gyrotrons.

