# OpenReview forum: "Multi-Timescale Dynamics Model Bayesian Optimization for Plasma Stabilization in Tokamaks"
_ICML.cc/2025/Conference — ICML 2025 poster_

### Official Review · Reviewer_j9tX · 2025-03-04

**Overall Recommendation:** 4

**Summary:**

The authors propose a principled approach using Bayesian optimization to optimize stabilizing actions to efficiently stabilize a tokamak (a type of nuclear fusion reactor). The author’s proposed approach integrates both more-reliable observed data from the tokamak when specific actions are taken, with less-reliable information from a data-driven dynamics model, into the GP surrogate model, in order to improve optimization performance. Experimental results demonstrate that the proposed approach improves tokamak stability over other approaches. Notably, the authors demonstrate that their proposed approach achieves a 117% improvement over past methods in avoiding tearing instability.

## update after rebuttal
As I stated in my comment below, the authors sufficiently answered all of my questions in their rebuttal, and I maintain that this work should be accepted for all of the reasons stated in my initial review.

**Claims And Evidence:**

Yes.

**Essential References Not Discussed:**

There are no missing references as far as I am aware. However, as I noted below, I am not at all familiar with existing literature on stabilizing Tokamaks or nuclear reactors in general, so if there is missing relevant work in this area, I might not be aware of it.

**Experimental Designs Or Analyses:**

Yes, I checked all experimental design setups and experimental results provided in the paper. All are valid as far as I am aware.

**Methods And Evaluation Criteria:**

Yes.

**Other Comments Or Suggestions:**

Typos:

1: In abstract: “Results on Live experiments…”. “Live” should not be capitalized here.

2: Near line 119-120: “... are kept fixed throughout each experiment roullout…”. “Roullout” should be “rollout”.

3: Near lines 156-158: “In the following sections, we individual components our method -”
Should instead be: “In the following sections, we discuss individual components of our method.”

4: Line 150: “measure the n=1 magnetic pertubations”, “pertubations” should be “perturbations”

5: Table 2 Row 7 “Decomponsed” should be “Decomposed”

6: Line 568 “suspectible” should be “susceptible”

**Other Strengths And Weaknesses:**

Strengths:

1: Application: This paper focuses on one specific application: stabilization of Tokamaks. This application is a strength of the paper because stabilizing these nuclear reactors is clearly an important and relevant real-world problem.

2: Compelling experimental results: The experimental results demonstrate a compelling improvement in stabilizing Tokamaks compared to past approaches. In particular, the 117% improvement in successfully avoiding tearing instability provides very compelling evidence that the author’s proposed method is better than past approaches for stabilizing Tokamaks.

Weaknesses:

The primary weakness of this paper is that there is a lack of novel machine learning methodology, as it combines ideas from well known methods in i.e. Bayesian optimization to create a principled approach for the specific problem of stabilizing Tokamaks. However, I think that the importance of the application and the compelling experimental results are strong enough that the novelty of the methods used are less important here. I therefore think that this is a very minor weakness and that the paper should likely be accepted.

**Questions For Authors:**

Question 1:
In your work on designing an approach specifically to improve tokamak optimization, what lessons have you learned that you think might be generally applicable to researchers trying to apply Bayesian optimization to other challenging problem domains? In particular, integrating high and low fidelity information in a single optimization run is a common problem in BO. In this case, the high-fidelity observations are the reliable ones from the Tokamak, and the low-fidelity/less-reliable information is that which we get from the dynamical model. What generally applicable lessons have you learned regarding this (if any)?

Question 2:
The RPNN uses a GRU cell to store information about past states and actions. What motivated this particular choice? It’s possible that performance of this model could be improved by using a few attention layers instead of a GRU. Using attention typically outperforms other approaches including LSTM, GRU, CNN, etc. (https://arxiv.org/abs/1706.03762).

**Relation To Broader Scientific Literature:**

This paper focuses on one specific application: stabilization of Tokamaks. Scientists who work with Tokamaks may be able to apply the proposed method in practice to more effectively stabilize these reactors. This application is relevant to the broader scientific community because nuclear reactor stabilization is a very relevant and challenging real-world problem.

**Theoretical Claims:**

There are no theoretical claims or proofs in this paper as far as I am aware.

---

> ### Author Rebuttal · Authors · 2025-04-01
>
> Thank you kindly for your detailed review. We have corrected all the typos you pointed out. Here are the answers to your questions:
>
> **Lessons:** During the experiments, we realized the importance of having a compressed representation of the actuator space. The high data efficiency of our approach would not have been possible if the search space had been much more granular. At he same time, having a high-fidelity model was particularly important, as we saw it clearly outperforms models with more naive and less sophisticated priors. During our experiment, we also observed that the suggestions for ECH provided by humans sometimes differed from our approach. We think investigating ways to incorporate these preferences on the fly would be a very exciting avenue of research.
>
>
> **Using self-attention:** We briefly experimented with an architecture that included self-attention while developing our model. Although we observed good predictive performance on the training data, the performance on the test data was poor, indicating overfitting. We speculate this is due to insufficient data and the overall tasks' difficulty (it has been shown that transformers sometimes only become good with a large amount of data (https://arxiv.org/abs/2106.04554)). For this reason, we stopped using transformers early during our research. Instead, we chose the RPNN architecture, which has been known to achieve state-of-the-art results on fusion data (https://arxiv.org/pdf/2404.12416). The paper cited looks into this question more deeply and analyzes GRU vs LSTM. Although we did not focus on this question in this work, we feel that designing appropriate attention-based architectures for nuclear fusion dynamics prediction is a very interesting research direction.

---

> > ### Comment · Reviewer_j9tX · 2025-04-01
> >
> > Thank for for answering both of my questions. I maintain that this work should be accepted.

---

### Official Review · Reviewer_GHhS · 2025-03-13

**Overall Recommendation:** 3

**Summary:**

This paper introduces a multi-scale Bayesian optimization approach, termed DynaBO, specifically designed to control tearing instabilities in tokamaks. The approach integrates a high-frequency Recurrent Probabilistic Neural Network (RPNN) with a low-frequency Gaussian Process (GP), allowing rapid adaptation between experiments. Offline experiments on historical data from the DIII-D tokamak demonstrated significant improvement over baselines, while live experiments at DIII-D resulted in a 50% success rate in suppressing instabilities, showing a 117% improvement over historical outcomes.

Strengths

The paper addresses a highly relevant, real-world application with significant practical implications for fusion energy.
The experiments conducted on the real DIII-D tokamak provide a compelling demonstration of the method's potential.

**Claims And Evidence:**

See above

**Essential References Not Discussed:**

See above

**Experimental Designs Or Analyses:**

See above

**Methods And Evaluation Criteria:**

See above

**Other Comments Or Suggestions:**

see above

**Other Strengths And Weaknesses:**

see above

**Questions For Authors:**

see above

**Relation To Broader Scientific Literature:**

See above

**Theoretical Claims:**

See above

---

> ### Author Rebuttal · Authors · 2025-04-01
>
> Thank you for reviewing our paper. If you feel that our responses to other reviewers improve the quality of the paper, we would be very thankful if you would consider changing your score accordingly.

---

### Official Review · Reviewer_zDdy · 2025-03-19

**Overall Recommendation:** 2

**Summary:**

- Machine learning algorithms face difficulties in controlling complex real-world systems, i.e., nuclear fusion.
- Existing methods like reinforcement learning and Bayesian optimization fall short of fully addressing these challenges.
- A new approach integrates a high-frequency data-driven model with a low-frequency Gaussian process to improve real-time adaptability.
- This method, validated on the DIII-D nuclear fusion plant, shows significant improvement over baseline approaches.
- Live experiments demonstrate a 50% success rate, marking a 117% improvement compared to historical results.

**Claims And Evidence:**

- It solves an interesting application of Bayesian optimization to nuclear fusion.
- Could you elaborate this sentence "a recurrent probabilistic neural network models the high-frequency model dynamics, while a Gaussian process models the effect of low-frequency marginal statistics on the dynamics"? In particular, what is the definition of frequencies here? What is the meaning of high and low frequencies?
- For the setting of a GP prior mean, can you compare your method to a GP prior mean with simple basic statistics.  For example, you can set it as the arithmetic mean of historical data.

**Essential References Not Discussed:**

This paper seems to discuss essential references.

**Experimental Designs Or Analyses:**

- In Figure 2, why did the authors vary a length scale?  Didn't the authors determine the length scale optimizing a GP model?
- In Figure 2, how did the authors choose a length scale for the Matern kernel?

**Methods And Evaluation Criteria:**

- How do the authors determine initials conditions and context?
- Details of the recurrent probabilistic neural network are missing.  For example, neural network architecture, training loss, and training scheme are missing.
- I don't understand Equation (5). How do you handle the Bernoulli distribution here?  Is it not a logistic function?
- Why is GP-UCB selected for an acquisition function?

**Other Comments Or Suggestions:**

- In Page 4, the authors argue "a Gaussian process (GP) model, a nonparametric model that is very data-efficient." I don't think GP is data-efficient.

**Other Strengths And Weaknesses:**

Please see above.

**Questions For Authors:**

Please see above.

**Relation To Broader Scientific Literature:**

It is related to broader scientific literature on Bayesian optimization.

**Theoretical Claims:**

It is not applicable for this work.

---

> ### Author Rebuttal · Authors · 2025-04-01
>
> Thank you kindly for your review. We have addressed your questions and comments in detail below.
>
> **Definition of frequencies:** By frequencies, we mean the resolution of the input signal. The RPNN takes as input the full step-by-step actuator signals, which have a frequency of 50Hz, and makes predictions on a recursive, step-by-step basis, at the same frequency. By contrast, the Gaussian process model takes only the average ECH and betaN as input, which corresponds to a low-frequency smoothed version of the ground-truth actuator signals, and outputs the residual of time to tearing mode. (In other words, the RPNN is a high-resolution model, whereas the Gaussian process is a low-resolution model.) Based on this discussion, we have substituted the sentence you indicated with a clearer one.
>
> **Comparison with GP with marginal statistics:** As you suggested, we have added the relevant comparison with a GP with marginal statistics as a prior mean. The table below shows the cumulative regret value after 500 iterations under the setup mentioned in the offline experiments setup. We observed that a prior mean corresponding to the historical mean of the data yields similar performance to that of a zero mean prior, indicating that a constant prior is insufficient for the present setting.
>
> || RBF ls.=0.01 | | RBF ls.=0.1 || RBF ls.=1 || Matern nu=0.5 | | Matern nu=2.5 |  | Linear ||
> | ----- | ----- | ----- | ---- | ---|----|----|-----|-----|-----| ------ | -------- | ------ |
> |  | Mean     | Std. | Mean     | Std.| Mean  | Std. | Mean  | Std.| Mean | Std. | Mean | Std.  |
> | Mean Prior  | 26899  | 30114.68 | 26971.6 | 30055.93 | 29228   | 28225.8  | 29478   | 28039.02 | 29417.8 | 28064    | 56309.4 | 6611.14  |
> | Zero Prior  | 27118 | 29939.2  | 27167.2 | 29895 | 30252 | 27432.5 | 29707.4 | 27845.35 | 29742.2 | 27798.6 | 51926.6 | 11134 |
>
> **Initial conditions and context:** The experiment configuration determines the initial conditions, which are agreed upon with DIII-D scientists before the experiment. The context for the GP is the target betaN, which is also decided before the experiment. We will make this clearer in the revised version.
>
> **Details of recurrent probabilistic neural network:**
> Network Architecture -
>
> Encoder :
>
> 1. FC layer (input dim x512)
> 2. FC layer (512x512)
>
> Memory Unit :
> 1. GRU Block (512 x 256)
>
> Decoder : Residual connections used between FC layers
> 1. FC Layer (256x512)
> 2. FC layer (512x512) repeated 8 times
> 3. FC layer (512x128)
>
> This connects to two outputs
> 1. Mean head (128 x output dim)
> 2. Log Var head (128 x output dim)
>
> The network predicts distribution parameters as output; hence, we train with a log likelihood loss. We use the Adam optimizer with a learning rate of 3e-4 and weight decay of 1e-3. We also use early stopping (patience = 250 epochs) on a validation set with 10% of the data points. We will include these details in the appendix of the revised manuscript.
>
> **Equation (5):** We train a binary classifier that outputs a probability of tearing mode at time t given the state s_t and action a_t. The output specifies a Bernoulli distribution (conditioned on s_t and a_t), which we sample to predict the probability of a tearing mode occurring. We will make this clearer in the revised manuscript.
>
> **Experiments with varying lengthscale:** The main goal of varying the lengthscale is to show that our method is robust to the choice kernel and hyperparameters, particularly compared to other approaches. We will state this more clearly in the revised manuscript.
>
> **Choice of acquisition function:** We ran more offline experiments with different acquisition functions:
> | Acq. fun. | RBF ls 0.1 || RBF ls 1 || Matern nu 2.5 ||
> | ----|---|-----|----|------|-----|-----|
> ||Mean| STD | Mean | STD | Mean | STD |
> | UCB | 14418.2    | 8038.94  | 10726.4  | 3717.18 | 10223 | 3997.86 |
> | Thompson Sampling | 11184      | 2835.95  | 15696.6 | 4091.07 | 18139.6 | 5818.07 |
> | EI | 8201 | 10183.31 | 10342.6 | 9859.99 | 10231.8 | 9582.02 |
>
> The numbers show the cumulative regret at the end of 500 iterations. We noticed similar performance across all acquisition functions. We chose UCB because it allows easy tuning of exploration vs exploitation during the actual experiment.
>
> **Lengthscale for Matern kernel:** As mentioned above, the offline experiments with varying lengthscales aim to analyze our approach's robustness under varying model specifications. With this in mind, we first observed the lengthscale obtained by log-likelihood maximization over the full data set, then chose lengthscales for the offline experiments that varied from the log-likelihood maximum by at most an order of magnitude. For the live experiments, we used the lengthscales that yielded the highest fit according to the log-likelihood loss.
>
> **GP data efficiency:** Thank you for pointing this out. We will revise the statement accordingly.

---

### Decision · Program_Chairs · 2025-05-01

**Decision:**

Accept (poster)

**Comment:**

This paper uses Bayesian optimization applied to stabilization in Tokamak nuclear reactors. Given the new "Application-Driven Machine Learning" track at ICML--to which this paper was submitted--I think this paper is clearly above the bar for acceptance under that track. As far as I can tell, there aren't substantial (or any?) criticisms leveled at the paper by the one rejecting review, and the two other reviewers are in favor. The one rejecting review hasn't updated since the author feedback period. The authors seem to get some pretty cool results here on a very important (from a societal perspective) application domain, and I'm not seeing an obvious reason resubmission would be necessary on this track.